# Goal-Directedness is in the Eye of the Beholder

## Abstract

Our ability to predict the behavior of complex agents turns on the attribution of goals. Probing for goal-directed behavior comes in two flavors: Behavioral and mechanistic. The former proposes that goal-directedness can be estimated through behavioral observation, whereas the latter attempts to probe for goals in internal model states. We work through the assumptions behind both approaches, identifying technical and conceptual problems that arise from formalizing goals in agent systems. We arrive at the perhaps surprising position that goal-directedness cannot be measured objectively. We outline new directions for modeling goal-directedness as an emergent property of dynamic, multi-agent systems.

## 1  Introduction

Selecting short-term actions to achieve long-term goals is central to human reasoning and intentionality [1]. As AI systems are being granted an increasing degree of autonomy, researchers have become interested in what it means for such agents to be goal-directed. Their approach has been largely *behavioral* [2, 3], claiming that we are justified in attributing mental states, such as intentions, where they are useful for explaining and predicting behavior. Others have adopted *mechanistic* approaches [4], which assume that intentions, or goals, correspond to distinct model states that can be measured by probing model internals.

The problem of detecting goal-directedness introduces several questions: *What exactly is a goal? How do we distinguish between having a goal and having the possibility of achieving it? How do we detect goal-directed behavior?* The core idea behind instrumentalist accounts of goal-directedness is that a goal, or the property of being directed toward it, is what causes the behavior that is associated with having that goal [5, 6]. An agent is defined as a decision-making system in an environment following specific objectives. The task of detecting goal-directedness in this way amounts to probing for the presence of unspecified objectives. The ability to monitor for the emergence of goals that might otherwise go undetected is understandably a key aim of AI alignment research.

In this paper, we complicate the story of goal-directed agents by working through the assumptions underlying behavioral and mechanistic approaches to goal-directedness [3, 4]. We first show a number of conceptual and technical problems with the definition in MacDermott et al. [3], as well as with related behavioral definitions. Their measure gives unintuitive results in pathological cases, and shows impossible to compute in others. We refer to such computability problems as measurement problems. Mechanistic accounts of goal-directedness also face demarcation problems. Xu and Rivera [4], for example, train classifiers on model activations from training with sparse versus dense loss functions, claiming that sparsity corresponds to goal-directedness. That is, from activations (model states) we can estimate whether a model is goal-directed or not. The demarcation problem is shared

between behavioral and mechanistic approaches: How do we distinguish between being directed toward one goal, and another that is specified[1] with a greater degree of granularity?

Both approaches come with ontological commitments. Behavioral measures depend on what is implicitly assumed in the underlying formalization of goals and agents, whereas mechanistic probes turn on the semantics of internal states. They also have assumptions in common: That goals are enumerable and can be specified in ways that make probing feasible. **We land on the position that goal-directedness cannot claim to be an objective measure.** Rather, it is only indicative of the fit between a formal model[2] and the system it is modeling. Taking cue from the biological literature on goal-directed organisms, we propose that goal-directedness research should not rely solely on anthropomorphic explanations, but should study how goal-directed behavior actually emerges in simulation. In §2, we provide background and preliminaries; in §3, we present the common challenges to behavioral definitions of goal-directedness, in §4, we turn to mechanistic definitions; and finally, in §5, we discuss possible implications and solutions.

## 2 Background

Both the mechanistic and behavioral approaches start out by asserting that an agent is best modeled as a node in a Bayesian Network (BN). The BN models the environment; the agent can, in theory, be a human, a non-human animal, a deep neural network or any other type of computer program. BNs are directed acyclic graphs (DAGs) modeling the dependence relations between probabilistic variables. Such networks have been used extensively as a formalism towards understanding inference and decision-making under uncertainty. Causal Bayesian Networks (CBNs) are BNs in which the graph edges encode not only dependencies, but represent causal relationships [7]. Causal queries are computed using intervention semantics, e.g., Pearl's do-operator [7]. The shift to CBNs was historically motivated by the observation that probability calculus is insufficient for knowledge-making of the kind that is important to science [8], e.g., the kind that show that disease causes symptoms, and not the other way around.

More recently, Everitt et al. [6] introduce Causal Influence Diagrams (CIDs); a formalism that modifies a CBN by decomposing the probability variables $V$ into random variables $X$, decision variables $D$, and utility variables $U$. Graphically, it extends a CBN with decision nodes (action choices, denoted as rectangular node) and utility nodes (agent preferences, denoted as diamond node). A CID is an extension of a CBN, in the same way that a traditional Influence Diagram (ID) is an extension of standard BNs. MacDermott et al. [3] adopt CIDs as the best formalization to model agent behavior, facilitating the quantification of goal-directedness. Goal-directedness is defined in the following way:

**Definition 2.1** (Goal-directedness [3]). A variable $D$ in a causal model is goal-directed with respect to a utility function $\mathcal{U}$ to the extent that the conditional probability distribution of $D$ is well-predicted by the hypothesis that $D$ is optimizing $\mathcal{U}$.

They illustrate the work the definition is supposed to do for us, through the familiar story of a mouse in a maze in search of cheese. In this story, we are met with a mouse in a grid world that may or may not have the goal of *eating cheese*. Typically, the mouse has to make a number of go-left-or-go-right-type decisions in order to get to the cheese. By Definition 2.1, we have reason to stipulate that the mouse has the goal of moving to where the cheese is, if its behavior ($D$) is well-predicted by the hypothesis that it is optimized for moving towards the cheese ($\mathcal{U}$). Goal-directedness is minimal when actions are chosen completely at random, and maximal when uniquely optimal actions are chosen. A mouse randomly walking about in the maze seems uninterested in cheese, but a mouse persistently moving in its direction seems set on it.

We will refer to the mouse-grid example throughout, but consider the parallel scenario in LLM safety research. Here, the goal of interest could be the LLM trying to prevent **sudo** access to its model weights, as well as preventing outside intervention in other ways. Consider the different components of the two thought experiments:

---

[1]For instance, winning a tennis match versus winning the same match within a margin, or in less than $n$ minutes.

[2]Here, we take the term formal model to mean the formalization adopted to model an agent making decisions in an environment.

| Agent | Goal | Environment |
|---|---|---|
| Mouse | Obtain cheese | Grid |
| LLM | Block **sudo** access | Server |
| $D$ | $\mathcal{U}$ | $X$ |

LLMs, briefly put, are functions $f(\cdot)$, with bells and whistles,[3] typically with billions of coefficients or weights. Since these weights are unfathomable to the engineer [9], it is customary to train linear and non-linear probes to probe for their capabilities and examine how they encode input internally.

Our main observation will be that what it means for an agent to have the intention of eating cheese, or revoking **sudo** access, is up for negotiation. This position is not merely one of linguistic relativism. Of course, the meaning of the word *intention* – or the meaning of the word *cheese*, for that matter – is under drift and continuously being negotiated by the linguistic community. What we are pointing to is deeper issue: Even if we stipulate a working definition of cheese, and a provisional concept of intention, we still face the question – what counts as *wanting* cheese? How bad do you need your want to be? Is wanting cheese tomorrow still wanting cheese? Is wanting cheese and olives, but *not* cheese on its own, an instance of wanting cheese? Is it possible to want cheese without being aware of it? These questions haven't been asked because they haven't mattered, until now. We propose that such conceptual ambiguities are not easily resolved, and for this reason, our operationalization of goal-directedness will have to be embedded in or take scope over simulations of social practices. We flesh out the argument for this position, as well as its implications for future research.

## 3 Behavioral Approaches

### 3.1 Syntactic Problems

The first class of problems have a syntactic or technical nature and could easily be addressed. The idea of defining goal-directedness relative to a goal-optimal model configuration runs into trouble when goals are beyond reach for models. Every agent has an inductive bias. Some agents are expressive, some are not. An LLM with a billion parameters can do more than a language model with five parameters. Some agents can model complex relationships; others cannot. In the limit, an agent can have no expressive power at all. We need to consider if the conditional probability distribution of a variable is well-predicted by the hypothesis that it is optimizing the utility function it is goal-directed towards. Meaning, we require that our measure can meaningfully express the distinction between being optimized toward a goal, and having the capacity to reach it. Several problems arise from the conflation of the two. Consider the following examples:

**Example 3.1** (No Cheese). Imagine a slightly modified version of the example in [3], in which the mouse still operates in a grid world, possibly looking for cheese, but in which there is no cheese. Since there is no cheese, there is no uniquely optimal strategy, or all strategies are optimal. Randomly walking about becomes indistinguishable from pursuing the goal of obtaining cheese.

The example shows how the behavioral definition of goal-directedness is too permissive, unless properly qualified. As it stands, any agent is goal-directed toward anything outside of its influence. There is another class of similar pathological examples that challenge the definition of goal-directedness in MacDermott et al. [3] in related ways. Consider the following example, which is not itself a challenge to MacDermott et al. [3], but an important stepping stone toward our second class of syntactic problems.

**Example 3.2** (The Cheese-Craving Stone). Imagine, again, a slightly modified version of the example in MacDermott et al. [3], in which the mouse has been replaced by a stone. Since the stone cannot move in any direction at all, random behavior again becomes indistinguishable from optimal behavior.

Proposition 3.3 [3] states that a system can never be goal-directed towards a utility function it cannot affect, and may thus already account for cheese-craving stones,[4] but what if we alter the example again?

---

[3]LLMs, as such, output probability distributions over next tokens. Bells and whistles are for sampling from these distributions to form coherent output.

[4]MacDermott et al. [3] derive their proof of Proposition 3.3 by showing that the maximum entropy goal-directedness of a mouse in a grid with no cheese, is 0. However, since 0 is the maximal value across all possible

**Example 3.3** (The Black Hole Collector). Imagine, again, a slightly modified version of the example in MacDermott et al. [3], in which the cheese is replaced with a black hole. Since the mouse moving in one direction or the other leads to the same result, i.e., the mouse ending up where the black hole is, random behavior becomes indistinguishable from optimal behavior.

Does Proposition 3.3 in MacDermott et al. [3] still save us? Maybe, but this depends on how we formalize things and what exactly is meant by affecting the utility function. We can certainly model the choices made by the mouse, leading to different states with the same utility. In other words, whether we think of a black hole-collecting mouse as goal-directed or not, depends on our underlying ontology.

Everitt et al. [10] have proposed another method of evaluating goal-directedness that attempts to distinguish goal-directed behavior from agent capability in task performance. Where we already know an agent has the relevant capability, we can observe how *willing* it is to use that capability towards a task. They first estimate the capabilities in controlled environments.[5] They then compute the optimal behavior given those capabilities. In theory, this approach could control for inductive biases and thus mitigate for the above pathologies, including the Cheese-Craving Stone and Black Hole Collector, in which case the optimal behavior will be severely limited by the inadequate capabilities of the agent.

## 3.2 Conceptual Problems

**Granularity**  Consider the ambiguity of the question whether the mouse has the goal of eating the cheese. Is the goal to eat the specific cheese, or will any cheese do? Could it be subsumed by the goal of staving off hunger? Would the mouse run after a new piece of cheese replacing the old one? Is the goal to eat the cheese right now or just to claim it now and eat it later? That is, if the cheese could only be eaten later, would the mouse still go for it? Is the goal to eat the cheese in its entirety or just sub-ingredients? If we split the cheese from its proteins, which part would the mouse go for? Would the mouse go for a piece of cheese if placed in another grid? And so on. It is trivial to complicate these examples beyond the toy example of a mouse in a grid. The general form of the problem is: How do we distinguish between the property of being directed toward the goal (environment state) described by propositions $S = \{p_1, \ldots, p_n\}$ and the property of being directed toward the goal described by propositions $S' = \{p_1, \ldots, p_{n+1}\}$? This turns out to be highly non-trivial to do in general, in the absence of precise definitions of the goals in question. Such definitions are highly impractical and may hinder generalization beyond toy examples.

**Uncertainties**  There is another form of conceptual problems, too: For each proposition $p_i$, how do we distinguish between $S$ and $S$ with $p_i$ replaced by $p_j$ with $p_i \rightarrow p_j$ or $p_i \sim p_j$? These problems are well-studied in logic and ontology [12]. What if we replaced the cheese with cream cheese or buffalo cheese? This is relevant for evaluating our measure of goal-directedness toward cheese, but also in a grid with several kinds of cheese, e.g., a grocery store. Entailments can also be derived from the relations: If the goal is *obtaining cheese*, for example, is the goal then satisfied by being granted the legal rights to the cheese? In real-life scenarios, such ambiguities compound.

## 3.3 Measurement Problems

CIDs are introduced as a formalism for modeling a *single* agent acting in an otherwise randomly distributed environment. This presumes that an agent's behavior is *uncaused*. that it's utility is unaffected by other agents' decisions. Yet in real-world, safety-critical settings, agents interact with humans [13] and other artificial agents [14]. Human goals are dynamically updated in response to shifting environmental, economic, and societal conditions [15]. To explore the feasibility of causal models in such contexts, we complicate the classic mouse-and-cheese example by introducing a second agent (Example 3.4).

**Example 3.4** (Two Mice). Two mice ($a$ and $b$) are placed in a grid with cheese at one end. Neither knows their position ($S_{a1}, S_{b1}$), but each can smell the cheese ($O_{a1}, O_{b1}$), observe the other's decision ($D_{a1}, D_{b1}$), and decide where to move. Their decisions are made simultaneously.

---

behaviors, this, technically speaking, does not just mean that a mouse in a grid with no cheese is *not* goal-directed, only that its maximum entropy goal-directedness is 0. In fact, all behaviors will be equally goal-directed toward the cheese in this case.

[5]This could maybe be done in a more general way by relying on so-called function vectors [11].

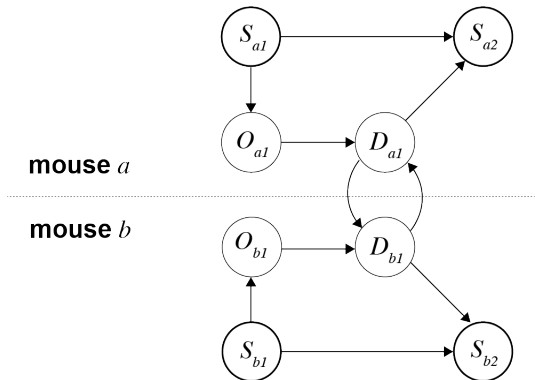

Figure 1: Example 3.4 modeled as a Causal Bayesian Network

This decision problem can be modeled with a CBN (Figure 1). The graph notably contains cycles, meaning that the joint distribution $P$ can no longer be factorized into conditional probabilities, and the problem as such is rendered computationally intractable. Multi Agent Influence Diagrams (MAIDs) have been developed to address such multi-agent dynamics by identifying equilibria where each agent maximizes expected utility [16], including cases with imperfect recall [17].

Example 3.4 can be reformulated as a cooperative or non-cooperative game. In the cooperative version, both mice benefit if the cheese is found, regardless of who reaches it. Meaning, mouse $a$'s utility is not dependent on the decision of mouse $b$ (Figure 1 b.). In the non-cooperative setup, we take it that the mice are competing to get to the cheese first. Moving simultaneously, $D_a$ is dependent on $D_b$, and each agent's utility node is affected by both decisions (their respective utility functions share the same parents), and so the relevance graph is cyclic. In fact, even in the case that mouse $b$ can first observe $D_a$, mouse $b$ must still know the decision rule of mouse $a$ in order to know how to proceed. For instance, if mouse $b$ observes mouse $a$ moving away from the cheese, $b$'s decision depends on determining whether $a$ is making a strategic bluff, or is simply bad at picking up scent. In such scenarios, strategies cannot be understood independently of recursive reasoning about the other agent's reasoning. Koller and Milch [16] propose a method to resolving cyclic dependencies in multi-agent settings by breaking the problem down into sub-games and calculating the Nash equilibrium for each in succession. Yet in practice, this problem scales exponentially with the number of possible decisions[6].

**Assumptions**  Below, we sketch out the branching assumptions involved in causal behavioral modeling. The first and most substantial of these is the assumption that an agent's utility function bears no causal relationship to the decisions made by other agents. This heuristic is what enables quantification of goal-directedness [3]. However, if the formalization adopted is insufficient to capture the decision problem we are claiming to model, then the resulting estimation of goal-directedness is bound to fail in predicting future behavior[7].

If instead we allow that one agent can be causally influenced by another, as in the minimal interactive structure of Example 3.4, then we are pushed toward game-theoretic frameworks in order to render the problem tractable. This forces us to assume either cooperative or non-cooperative strategy structures, alongside familiar assumptions in game theory (such as perfect information and common knowledge), we are also limiting the space of possible intentions to a highly restricted class of strategic forms.

Interactive PODMAPS (I-PODMAPS) and their graphical counterparts (Interactive influence diagrams (I-DIDs) [19]) present an alternative to game-theoretic modeling, which adopts the perspective of a single agent, inferring the beliefs of the second. The key departure of I-DIDs from MAIDs is the inclusion of a model node which contains the candidate models of the second agent, in the most general sense. However, I-PODMAPs notably suffer from the *curse of dimensionality*, as the

---

[6]Hammond et al. [18] propose a method of equilibrium refinement in which the cyclic component of the graph is collapsed into a single node, which is represented and solved as an Extensive Form Game (EFG). Yet, the resulting EFG problem also grows exponentially with the size of the strategy space.

[7]Looking again at Example 3.4. If mouse $a$ assumes that mouse $b$ *cannot* be influenced by its own actions, then $a$ is missing a crucial aspect of reasoning.

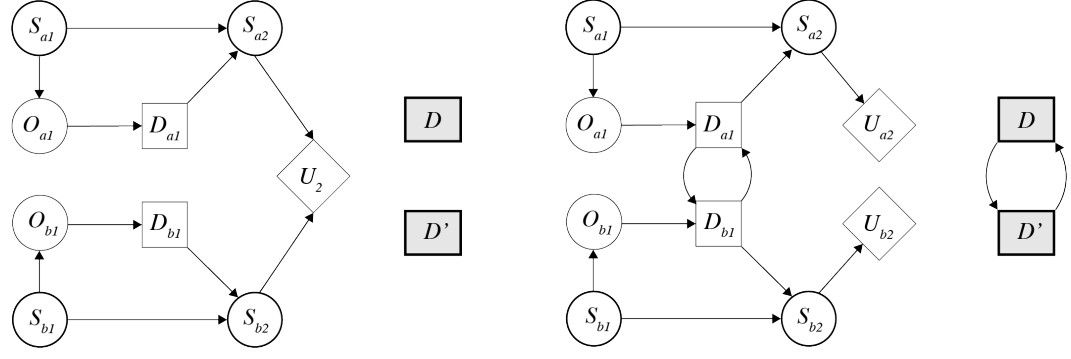

a) Causal Influence Diagram (Cooperative)    a) Causal Influence Diagram (Non-Cooperative)

Figure 2: Example 3.4 represented as a CID for a cooperative (left) and non-cooperative (right) game, along with the associated relevance graph

interactive state space encompasses both observed behavior, and the space of candidate models[8]. Interestingly, the need for heuristics and approximations points towards a more pervasive problem in the causal modeling of agent decision-making. Namely, one of recursion in the modeling of another agent's beliefs [20]. Intentional modeling inevitably involves modeling an agent who in turn is modeling the second who is in turn modeling the first. The depth of recursion presents as a computational limitation, which is reflected in the literature on human cognition[9] as bounded rationality [22].

What does this mean for measuring goal-directedness? It suggests that accounting for mutual influence between agents renders the modeling of goal-directedness computationally intractable. This raises a deeper question: If a phenomenon resists formal measurement within a given model, does that imply it is absent? Or merely that the model's assumptions are insufficiently expressive? Absence of measurement isn't evidence of absence, but it might be evidence of an inadequate modeling framework. We suggest that a possible direction for future research in goal-directedness might begin with questioning the foundational assumption that goal-directed behaviour is best modeled in a bottom-up manner, with internal goals as the cause of observed behaviour.

# 4 Mechanistic Approaches

## 4.1 Conceptual Problems

MacDermott et al. [3], among others, have relied on instrumentalist accounts of goal-directedness. However, explaining behavior by appealing to optimal strategy is often neither computationally possible nor meaningful. One reason for the latter is that any departure from the optimal strategy in parameter space can be almost arbitrarily far from the target goal in human, conceptual space. The alternative is to take a more mechanistic approach, looking at the internals, as proposed by Xu and Rivera [4]. While mechanistic accounts face their own conceptual problems, they do seem to resolve some of the problems of behavioral accounts. The behavioral account turns on our specific definitions of goals and agents[10]. Mechanistic accounts instead sample common examples of systems directed towards goals, and hope the probe learns to generalize from them. Of course the lack of exact criteria for being directed toward a goal will compromise our ability to evaluate for robustness. More importantly, however, mechanistic accounts stir up new conceptual problems.

**Multiple Realizability**    Behavioral accounts black-box systems and need not worry about the possibility of multiple realizability. Being goal-directed toward cheese may look the same across systems, while being implemented in radically different ways. What it looks like for one system to be

---

[8]This intractability is further exacerbated by the depth of recursion, as well as depth in time.

[9]Humans of course face cognitive limitations when it comes to recursive reasoning, and have been shown to not engage in nested reasoning beyond two or three levels of depth [21].

[10]Including the formal models employed along the way

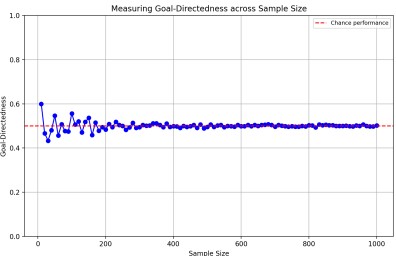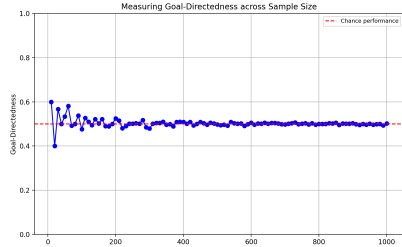

Figure 3: Goal-directedness is not learnable for linear (left) or non-linear (right) probing classifiers.

directed toward cheese, may be different from what it looks like for another system. Even within a single system there may be multiple algorithms implementing goal-directedness toward cheese. This poses a serious challenge to a probe based exclusively on internal states.

**Externalism**   A more subtle challenge is that goals need not always be fully internalized. To see this, imagine a mouse in a grid that learns to search for cheese, but is only aware of its search for something yellow. The mouse does not need to have an awareness of the goal in its entirety, in order to be directed towards it. Or a therapist explaining to her client that what she is really searching for, is recognition. Or an astronomer explaining to the astronaut that she is not really on her way to the Evening Star, but to Venus. The general point, it seems, is that an agent's goal need not always be completely encoded in its internals. A goal is in part defined by the external environment.

## 4.2   Measurement Problems

Probing for goal-directedness by probing internal model states only makes sense if we assume that we can detect traces of the optimal strategy directly in model parameters. In other words, it turns on an essentialist assumption that there is something to model. This runs up against the idea of multiple realizability, and it is fairly easy to show the inefficacy of this approach in practice.

To do so, we trained up to 1,000 linear feed-forward neural networks on one of two different tasks or goals. In both cases, we set up the tasks so that they were linearly separable, guaranteeing convergence. We then passed on the 1,000 induced classifiers to a goal-probing classifier. We experimented with both linear and non-linear probing classifiers. Their input was the raw model weights, and we evaluate classifiers by using cross-validation over random splits. The two tasks were synthetically generated to be different, sampling data points from two distinct pairs of Gaussians with different means and variances.

Figure 3 illustrates how the induced goals are clearly not learnable. As soon as we have statistical support, results coincide with chance performance. This may of course be due to the inductive bias of the learning classifier, but we see the exact same behavior for both linear and non-linear probes. We argue that there is a deeper reason for our failure to induce these goals. Goals are not directly encoded. Or, in other words, goals do not have unique keys in discriminative classifiers. For most problems, the goal is multiply realizable to the extent that most pairs of goals become indistinguishable.

## 5   Discussion

**Measurement Problems**   Dennett's instrumentalist account of intentionality [23] has been influential within the AI community, but we argue that mechanistic approaches are more aligned with the Belief Desire Intention (BDI) frameworks in philosophy of action [24]. Where the latter presumes a causal relationship between an agent's internal state and their resulting action (i.e. a reason for acting), the former does not. Instrumentalism embeds intentionality as simply one level of explanation that can be called upon whenever a system is too complex to warrant a physical or design level account [25]. In the standard BDI frameworks, reason and action are exclusive properties of an agent. Under the intentional stance, a reason is the best explanation that one (or another) could give for an action.

Intentional attribution is pluralistic and context-dependent. Multiple, equally valid intentional interpretations can coexist if they each yield successful predictions in their respective contexts. Motivating goal-directedness on this account means outright foregoing the possibility of objective measurement. What is it that we claim to measure then? The proposed measurements cannot be said to track goals, otherwise we find ourselves inadvertently sneaking in essentialism again. This is not to say that measurement itself is a misguided effort. Rather, simply to acknowledge the tension between instrumentalist paradigms and the ontological commitments that measurement often brings with it. In this case, goal-directedness measures should be regarded as just another observation. They cannot be said to reveal an objective, underlying property of the system in question. Rather, the measurement is revealing only of the relation between a system and the modeling framework used to observe it. Measurement as such is dependent on the instruments used. If we probe for intentional behavior, we will of course find instances of it.

**Goals in Biological Systems**   Intentional attribution allows us to predict animal behavior, but it doesn't establish whether animals actually have intentions. Heyes and Dickinson [26], for instance, argue that intentionality in non-human animals can only be tested under strict lab conditions, implying that behaviors like approaching food are not inherently intentional. Much of the discussion and relevant work in biology (see Allen and Bekoff [27]), runs into the same conceptual problems of goal specificity[11] as laid out in Section 3.2.

Early reliance on anthropomorphic interpretations of biological organisms often obscured underlying mechanisms. While attributing intentionality can aid heuristic understanding [28], mechanistic accounts have explained goal-directed behavior in organisms such as planaria, bacteria, or regenerating tissues without invoking intention or representation [29]. For instance, El-Gaby et al. [30] found a biological correlate for goal-directed behavior in mice that is crucially not defined in terms of optimal policy. Rather, they find that *goal-progress* is learned as a general task structure encoded at each behavioral step. That is, the mice do not need to represent a goal explicitly in order to reach it. They instead represent their progress within a task structure that directs behavior towards several possible outcomes. Hill et al. [31] similarly defend the view that goal-directed behavior is not caused by specific goals or environmental states, as per the standard account, but "normative patterns of action".

This literature informs how we are to understand goal-directedness of AI agents. Biological organisms learn how to behave in a goal-directed manner, but not with a particular goal in mind. Rather, what they learn is how to traverse structured environments predictably. It goes without saying, biological and artificial agents are not the same. Yet, if we are to borrow a concept from biology, it might also be wise to adopt the philosophical ambiguity that surrounds it.[12] In light of this research, we can see how existing mechanistic approaches may search in vain for goal-directedness towards specific goals. This is because there need not be a representation of the goal itself. Behavioral accounts are also challenged, for if goal-directed behavior is the result of a local, step-wise optimization process, there is no guarantee that goal-directedness is optimal over the full trajectory.

**Simulating Goal-Directedness**   One of the key motivations for probing agents for unspecified goals is to ensure safe deployment of AI systems. How can we monitor whether agents are developing instrumental goals that might lead systems or subsystems to inflict harm on our fellow humans? Can we monitor the safety of agentive systems in the absence of intentional attribution? One approach to monitoring safety is simply 'rolling the tape', i.e., observing its real-life behavior. Of course if the system is dangerous, rolling *any* tape would be irresponsible. However, just as is the case with humans learning to fly airplanes, the solution is to roll the tape in controlled environments: computer simulations or real-life role plays.

What would a controlled environment look like, and what observations would guarantee the safety of an AI system? Piatti et al. [32] evaluate the capability for collaboration of reasoning models in synthetic game scenarios. The relevance of such simulations turns on how well real-life scenarios

---

[11]How do we know whether a biological mouse wants cheese, mozzarella cheese or just that brand of mozzarella cheese? How do we know whether it wants cheese in general, or just here and now? How do we know if it eats the cheese to satisfy hunger, or to prevent anyone else from eating it? And so on.

[12]Hill et al. [31] argue that conflating goal-directedness with its putative explanation risks collapsing the descriptive and explanatory projects into one. Meaning, goal-directedness can and should be understood as a phenomenon independent of its utility in explanation.

have been simulated, as well as how trivial or non-trivial it would be to mitigate potential harm. Sullivan [33] has discussed both aspects under the heading of *link uncertainty*.

Recent work has analyzed the reasoning logs of LLM agents to show that they can exhibit goal formation that deviates from their explicit instructions [34, 35]. These approaches monitor misalignment without measuring goal-directedness across the action space—nor do they turn on our ability to probe internal states. Do these qualify as instances of "rolling the tape"? Perhaps, but their usefulness hinges on how likely such behaviors are to arise in real-world contexts, and whether they would plausibly lead to harm. The link uncertainty is, in other words, high in such studies. Moreover, manual analysis of reasoning logs introduces a high degree of subjectivity. It is also rather cumbersome in its reliance on human annotators, and yet, real impact on end users is not measured, only impact *imagined* by the annotators. This is why, instead of human analysis of reasoning logs, we propose to evaluate the goal-directedness in context, in a realistic simulation of agents acting within and upon an environment.

Importantly, simulations do not require supposing mental states such as intentionality. Rather than attempting to detect or define goals, simulations can be used to observe how patterns of behavior unfold under varying constraints. Because simulation tracks behavior over time and, crucially, *in context*, we can examine features of goal-directedness (e.g. persistence, norm-sensitivity, or causal intervention) without appealing to anthropomorphism. We can then ask: How does goal-directed behavior arise in AI systems? Taking cue from biological literature, we propose a treatment of goal-directedness as a phenomenon that precedes its role in explanation.

## 6   Alternative Views

Our position stands that the attribution of goals is conceptually slippy, runs into measurement problems, and cannot be directly probed for. This is in opposition with the prevailing view that identifiable goals can be encoded in an agents internals. Many researchers continue under the assumption that this view is correct. Their position would be to accept that goal-directedness is elusive *in theory*, but still has practical value; that the assumptions made about the nature of goals and agents are just useful heuristics; that goal-directedness measures simply serve as another tool in the toolbox. We are amenable to this position, and do not claim that the measures are fundamentally misguided. However, we do suggest that the assumptions of the modeling frameworks are foregrounded, and the application of such methods is limited to appropriate settings.

A third position would be to agree with our skepticism around quantifying goal-directedness, but suggest a solution other than simulation – or to argue there is no solution at all. We welcome alternative solutions, and note one convincing argument against simulations: The population that we are trying to model through simulation is under constant drift. We can run simulations familiar to us according to how LLM agents are used in practice, but for our simulations to be relevant down the road, we would, in theory, need to predict how LLM agents might be used in the future.

## 7   Conclusion

Proposed methods for measuring goal-directedness rely on implicit assumptions that fail to generalize to complex real-world settings. Behavioral methods turn on our precise definitions of both *goals* and *agents*. For the former, we quickly run into insurmountable conceptual ambiguities. For the latter, CIDs are adopted to model agent behavior, however they fail to model complexity beyond toy examples. The heuristics that make such methods tractable are also what severely limit their scope. A mechanistic approach does not turn on such definitions, but it does assume that goals can be learned and embedded in internal model states in ways that make them accessible to probing. Both approaches risk reifying an internalist conception of goals, undermining the instrumentalist argument that they are founded upon.

We propose that goals need not be intrinsic properties of agents. Limiting goal-directedness to what can be internally specified risks missing the broader dynamics at play. Namely, we require methods that can model goal-directed behavior without explicit, internalist goal representation, and instead as behavior that emerges through dynamic interaction with the environment. To this end, we suggest multi-agent simulation as a suitable methodological approach for identifying and diagnosing the conditions under which goal-directed behavior emerges.

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
