# OpenReview forum: "Goal-Directedness is in the Eye of the Beholder"
_NeurIPS.cc/2025/Position_Paper_Track — Submitted to NeurIPS 2025 Position Paper Track_

### Official Review · Reviewer_WCQ8 · 2025-08-14

**Significance:** 2
**Presentation:** 3
**Rating:** 6
**Confidence:** 2

**Summary:**

This paper presents a critical analysis of current methods for identifying goal-directed behavior in AI agents. They argue that goal-directedness is an attribution by an observer. They examine behavioral and mechanistic approaches.

The paper argues that both approaches suffer from significant technical and conceptual problems. Behavioral methods face "syntactic problems" in pathological cases (e.g., when a goal is unreachable) and "conceptual problems" regarding the ambiguous granularity of goals (e.g., wanting cheese vs. wanting to stave off hunger). These methods also become computationally intractable in multi-agent scenarios. Mechanistic approaches are challenged by the principles of multiple realizability (a single goal can be implemented in many ways) and externalism (a goal may not be fully encoded in an agent's internal states). The authors support this with a simple experiment showing that linear and non-linear probes fail to distinguish between agents trained on two distinct goals.

**Strengths:**

1. There is a strong critical analysis provided, revealing significant and overlooked conceptual ambiguities.
2. Effective use of examples
3. Concepts from philosophy of mind, CS are thoughtfully integrated

**Weaknesses:**

1. The multi-agent simulation alternative is not fully fleshed out.
2. It is unclear if this result would hold against the more sophisticated models (e.g., Transformers) currently used in the field.
3.  An alternative view, which the paper acknowledges but could engage with more deeply, is that current measures are useful, if imperfect, proxies necessary for practical safety engineering.

**Questions:**

1. Your paper argues that goal-directedness is in the "eye of the beholder." How does your proposed solution of using simulation escape this critique? Doesn't an observer still need to interpret the simulation's results and attribute goals to the observed emergent behaviors?
2. You raise excellent points about the ambiguity of a goal's granularity (e.g., "obtain cheese" vs. "stave off hunger"). How would a simulation-based approach be designed to distinguish between these different levels of abstraction without an observer imposing them?

**Alternative Position:**

Yes, and alternative positions are trivial straw-man arguments

**Author Identification:**

No.

**Context:**

3

**Discussion:**

3

**Ethics:**

["NO or VERY MINOR ethics concerns only"]

**Position:**

Yes, the paper argues for or against a position related to machine learning.

**Support:**

3

**Thoroughness:**

4

---

### Official Review · Reviewer_mMFX · 2025-08-15

**Significance:** 3
**Presentation:** 3
**Rating:** 6
**Confidence:** 2

**Summary:**

The paper provides a position about goal-directedness, in particular, the position says that goal-directedness cannot be measure objectively. The paper considers two types of approaches towards goal-directedness: behavioural and mechanistic, then provide various arguments as well as examples to derive towards the proposed position, which is that goal-directedness is not enumerable.

**Strengths:**

+ The paper aims to propose a proposition in an important research area, which is to consider goal-directed behaviour of agents in complex environments.
+ The paper writing is really good. I really enjoy reading this paper. The paper provides a very clear proposition as well as the supporting arguments.
+ The paper also provides various related works as well as various related positions.

**Weaknesses:**

I think I would like more if some technical details were explained in a clearer way in the paper.

+ First, the definition of goal-directedness should be explained clearer, and also, the scope of the types of agents the paper considers is not clear. The paper states that an agent is modelled as a node in a Bayesian Network; is this true for all types of agents? If not, then the paper needs to make this clear.

+ Second, the arguments for the position, even very well-explained, are quite intuitive and general. For me, I hope for a more technical arguments for the position.

+ Third, just my point of view regarding the position, I’m just wondering if the reason the goal-directedness cannot be measured objectively is that we do not have a well-defined definition of goal-directedness. If we have this ideal definition, can goal-directedness be measured appropriately?

**Questions:**

The authors could answer the comments and questions I listed in the Weaknesses section.

**Alternative Position:**

Yes, and alternative positions are well-considered and addressed by the argument

**Author Identification:**

No.

**Context:**

3

**Discussion:**

3

**Ethics:**

["NO or VERY MINOR ethics concerns only"]

**Position:**

Yes, the paper argues for or against a position related to machine learning.

**Support:**

2

**Thoroughness:**

4

---

### Official Review · Reviewer_sRRL · 2025-08-29

**Significance:** 3
**Presentation:** 2
**Rating:** 5
**Confidence:** 2

**Summary:**

This paper discusses goal-directed behavior. It claims that goal-directedness cannot be measured objectively. Then, it outlines new directions for modeling goal-directedness as an emergent property of dynamic, multi-agent systems.

**Strengths:**

1. The problem studied is important and fundamental.

2. The writing is very good. It provides sufficient background knowledge, shows evidence regarding the limitation, and then points out new directions.

**Weaknesses:**

I am not familiar with this area. So my question might be very superficial.

1. The conclusion is obtained from a toy example under some assumptions. Does the assumption hold for practical applications? Does the obtained conclusion work for the generic settings? If not, the foundation of this position paper is problematic. It would be good to provide more convincing results.

2. As a position paper, it would be good to provide more discussions for the future direction. The current version just uses two pages to point out the future direction. It is kind of not sufficient. It would be good to provide more comprehensive discussions for the future direction.

**Questions:**

1. The conclusion is obtained from a toy example under some assumptions. Does the assumption hold for practical applications? Does the obtained conclusion work for the generic settings? If not, the foundation of this position paper is problematic. It would be good to provide more convincing results.

2. As a position paper, it would be good to provide more discussions for the future direction. The current version just uses two pages to point out the future direction. It is kind of not sufficient. It would be good to provide more comprehensive discussions for the future direction.

**Alternative Position:**

Yes, and alternative positions are well-considered and named but not addressed

**Author Identification:**

No.

**Context:**

3

**Discussion:**

3

**Ethics:**

["NO or VERY MINOR ethics concerns only"]

**Position:**

Yes, the paper argues for or against a position related to machine learning.

**Support:**

2

**Thoroughness:**

3

---

### Note · Authors · 2025-09-04

**1-11 Submit Again:**

Definitely yes

**1-1 Submission Process:**

4

**1-2 Next Year:**

Perhaps a rebuttal phase as in the main track.

**1-3 Future Development:**

Loosening the constraints on the style/structure of the papers allowed.

**1-4 Interest:**

["Panel discussions with other position paper authors", "Mentorship programs for early-career researchers"]

**1-4 Other Interest:**

A dedicated position paper stream.

**1-5 Thoughtful:**

7

**1-6 Supportive:**

9

**1-7 Technical Aspects Versus Position:**

4

**1-8 Gate Keeping:**

10

**1-9 Camera Ready Changes:**

Section 2. Clarify the definition/formulation of goal-directedness we are interested in, and how it relates to the broader concept in biological literature.
Section 3.3. Add a formal proof of the claims in section 3.3 in the appendix.
Section 4.2. Expand further around how other model architectures would fare in detecting learned goals, in response to weakness raised by reviewer WCQ8. (We concede that multi-layer perceptrons are not Turing complete. It is therefore, strictly speaking, an empirical question as to whether other architectures could do a better job. However, we're skeptical that even RNNs with unbounded memory would do better. This is because of the inductive bias of feed-forward networks and their lack of internal structure. If goals have footprints, these should be the kind of footprint detectable by a multi-layer perceptron. )
Section 5. Outline in which contexts such goal-directedness measures might still be useful and complementary to proposed alternatives (in response to point raised by reviewer WCQ8)
Section 5. Clarify how the proposed alternative of simulation can be used to identify and anticipate possible undesired goal-directed behaviour (for AI safety research) without the need for ascribing or looking for specific goals.
Section 5. Offer further justification for the simulation proposal as supported by existing work in the field that adopts such a methodology to related problems such as concept emergence and drift in LLMs. Otherwise, clarify that the fleshing out of an alternative methodological approach is left for a constructive, technical paper.

**3-1 Review Response1:**

mMFX

**3-2 Reaction To Review1:**

"First, the definition of goal-directedness should be explained clearer, and also, the scope of the types of agents the paper considers is not clear. The paper states that an agent is modelled as a node in a Bayesian Network; is this true for all types of agents? If not, then the paper needs to make this clear."
Our paper is responding to the AI safety literature which has proposed that it's useful to model agents as (uncaused) nodes in a bayesian graph. Agents and agency can of course be modeled using other formalisms/methods, which is precisely what we suggest as a finding of our analysis. We will clarify this point, as well as expand upon the particular definition of goal-directedness we are referring to here.

"Second, the arguments for the position, even very well-explained, are quite intuitive and general. For me, I hope for a more technical arguments for the position."
For the computational intractability claims in section 3.3, we will add a formal proof in the appendix.

"Third, just my point of view regarding the position, I’m just wondering if the reason the goal-directedness cannot be measured objectively is that we do not have a well-defined definition of goal-directedness. If we have this ideal definition, can goal-directedness be measured appropriately?"
MacDermott et. al do provide a very precise definition of goal-directedness with respect to their modeling framework (causal influence diagrams). We argue that goal-directedness definitions and measures will always rely on the chosen modeling framework, which introduces an element of subjectivity to the method. Just as in Dennett’s view of intentional attribution, goal-directedness is subjective insofar as it can only reflect the relationship between system and observer (modeling framework) rather than an inherent property of the system itself. This is a nuanced point that we believe has thus far been overlooked in goal-directedness research.

**3-3 Review Response2:**

WCQ8

**3-4 Reaction To Review2:**

We agree with WCQ8 that our proposed alternative is not fully fleshed out, however the primary contribution of this position paper is to critique and explicate baked-in philosophical assumptions in goal-directedness measures, identifying possible pitfalls of the approach. We argue that multi-agent simulation bypasses some of these pitfalls by doing away with the need for goal attribution whatsoever. In other words we are laying the groundwork for an alternative, and complementary, approach, as motivated by the argument presented. The fleshing out and testing of an alternative is left to a technical paper, and would not be appropriate for the position paper track.

"Your paper argues that goal-directedness is in the "eye of the beholder." How does your proposed solution of using simulation escape this critique? Doesn't an observer still need to interpret the simulation's results and attribute goals to the observed emergent behaviors?"
We suggest multi-agent simulation as an alternative for the purpose of identifying potential risks from AI - however we argue that this can in fact be done without appealing to the concept of goals or measurement of goals. That is, without ascribing intentions on the Dennett account, but rather letting the behaviour of the simulated system play out. The benefit of simulation is that we may capture emergent, or computationally irreducible high-level phenomena that might otherwise be missed. We will further clarify this point in the Discussion section.

To the final question, the proposed method of simulation would not attempt to identify goals, but play out the course of action in a controlled setting, and observe what actually took place. Although the analysis of the simulation results requires some abstraction as well, it does bypass the hypothesising of particular goals. From here, we are better equipped to identify risks in AI systems, as well as the underlying interactional dynamics that led to them.

**3-5 Review Response3:**

sRRL

**3-6 Reaction To Review3:**

"The conclusion is obtained from a toy example under some assumptions. Does the assumption hold for practical applications? Does the obtained conclusion work for the generic settings? If not, the foundation of this position paper is problematic. It would be good to provide more convincing results."
Perhaps there is some confusion here - Our analysis is specifically criticising the toy example that is used in the existing work on quantifying goal-directedness. Our paper unpacks the underlying assumptions baked into this toy example, in order to argue that the resulting measure (i.e., MEG) is highly limited in practical, real-world applications.

"As a position paper, it would be good to provide more discussions for the future direction. The current version just uses two pages to point out the future direction. It is kind of not sufficient. It would be good to provide more comprehensive discussions for the future direction."
As already noted, this paper sets out to offer a critical analysis of existing approaches, and to point towards alternatives, however we leave the constructive contribution of simulating the emergence of goal-directed behavior for a technical paper. Although we find two pages of future direction to be sufficient, we agree that the proposal can be further clarified and supported with reference to existing work in the space, which we can add for the camera-ready.

---

### Meta-Review · Area_Chair_16Me · 2025-09-06

**Rating:** 6
**Confidence:** 3

**Strengths:**

– Important problem/area
– Presentation
– Provides ample context (related work and positions)
– Strong critical analysis

**Weaknesses:**

– Definition of goal-directedness (a fundamental concept in the paper) is not adequately presented.
– Lack of depth in arguments supporting the position.
– Unclear what the results would be when using stronger models.
– Lack of discussion of some of the alternative views.

**Questions:**

– R1’s questions regarding the foundation on a simple scenario.
– R2’s questions regarding the weaknesses they point out.
– R3’s questions regarding how the authors proposal would address the issues raised in the paper.

**Ethics:**

No.

**Thoroughness:**

3

---

### Decision · Program_Chairs · 2025-09-26

Reject